# Job satisfaction of clinical pharmacists and clinical pharmacy activities implemented at Ho Chi Minh city, Vietnam

Hai-Yen Nguyen-Thi[1]*, Thuy-Tram Nguyen-Ngoc[1], Minh-Thu Do-Tran[1‡], Dung Van Do[2‡], Luyen Dinh Pham[1‡], Nguyen Dang Tu Le[1]

1 Pharmaceutical Administration Department, Faculty of Pharmacy, University of Medicine and Pharmacy at Ho Chi Minh City, Ho Chi Minh City, Vietnam, 2 Ho Chi Minh City Department of Health, Ho Chi Minh City, Vietnam

☉ These authors contributed equally to this work.
‡ These authors also contributed equally to this work.
* haiyen@ump.edu.vn

**Data Availability Statement:** Data are available at https://figshare.com/articles/dataset/Job_satisfaction_of_clinical_pharmacists_and_clinical_

## Abstract

Ho Chi Minh City (HCMC) in Vietnam pioneered the practice of clinical pharmacy; however, hospitals in HCMC have faced numerous challenges that might influence the job satisfaction of clinical pharmacists (CPs). Additionally, there have been no official statistics about clinical pharmacy activities that have been reported so far. Therefore, this study was conducted to examine the current status of the clinical pharmacy profession and to analyze the key factors affecting job satisfaction of CPs in HCMC. This was a cross-sectional study. Self-administered questionnaires were distributed to all the CPs in all the 128 hospitals in HCMC via an online survey tool from May to June 2020. Only about 30% of the respondents were full-time CPs. The percentage of CPs participating in clinical wards was relatively low (52.79%). *"Provide drug information for patients and medical employees"* was the most common clinical pharmacy activity, with the percentage of CPs participating in it being nearly 90%. Overall, 74.1% of the 197 CPs surveyed were satisfied with their current job. The factors that they were satisfied with the most and the least were *"Inter & Intra professional relationships"* (95.9%) and *"Income"* (59.9%), respectively. The only demographic and work-related characteristic that had a statistically significant association with overall job satisfaction was *"Ward round participation"*. Most clinical pharmacy tasks noted a high rate of participation from the CPs. Nevertheless, hospitals in HCMC was found to be experiencing a shortage of CPs and low levels of participation of CPs in ward rounds, and most CPs were unable to completely focus on clinical pharmacy tasks. Regarding CPs' job satisfaction-related aspects, income and ward round participation appear to be the two factors that should be increased, in order to enhance CPs' job satisfaction.

## Introduction

Clinical pharmacy is a health science discipline and a patient-oriented service that has been developed to foster the rational use of medicine [1]. It has been a common and core practice

pharmacy_activities_implemented_at_Ho_Chi_
Minh_city_Vietnam/13549679.

**Funding:** The authors received no specific funding
for this work.

**Competing interests:** The authors have declared
that no competing interests exist.

within healthcare systems in developed countries since the 1960s [2, 3]. However, it was not until 2012 that the Vietnamese Ministry of Health (MOH) imposed the first ever regulation to determine and elaborate on the role and tasks of a clinical pharmacist (CP) *(Circular No. 31/ 2012/TT-BYT)* [4]. This legal document came into existence because clinical pharmacy was into being practiced at a few top-ranked hospitals via international collaborative programs, and not at many other small and medium-sized hospitals. Thus, there was a necessity to promulgate an official legislation to guide and to oblige all hospitals in Vietnam to gradually implement clinical pharmacy services in their hospitals. The relatively late implementation of clinical pharmacy services in healthcare practice in Vietnam results from the presence of various barriers and limitations, such as shortage of workforce, the lack of qualified clinical pharmacists (CPs), and the lack of interaction between CPs and other healthcare professionals [5–7].

Ho Chi Minh City (HCMC), which is situated in the southeastern region of Vietnam, is one of the biggest healthcare centers in Vietnam with a huge number of inpatient and outpatient visits on a daily basis [8]. As a pioneer in implementing clinical pharmacy services, HCMC has undoubtedly witnessed even bigger challenges in introducing and combining clinical pharmacy with other conventional pharmacy practices within hospitals [9]. More specifically, a study conducted in 2019 [5] showed that most hospitals in HCMC failed to meet the standard established by the Vietnamese MOH in terms of the quality and quantity of the clinical pharmacy workforce. Furthermore, most CPs had to simultaneously handle numerous traditional pharmacy tasks, such as drug dispensing, compounding, and administrative tasks, besides clinical pharmacy activities [5]. All these reasons are likely to result in low job satisfaction among CPs which further reduces job performance and indirectly harms patients [10–12]. Thus, developing policies that aim to enhance job satisfaction for CPs should be high on hospitals' agendas, in order to more effectively attract and retain CPs. However, since Circular 31/2012/ TT-BYT took effect, no official statistics have compiled the clinical pharmacy activities that are being practiced at healthcare facilities in HCMC. For these reasons, this study was carried out to examine the current status of the clinical pharmacy profession and to analyze the key factors affecting job satisfaction of CPs.

## Methods

### Study design

This study was designed as a cross-sectional study. Data used in this survey was gathered through an online survey conducted from May to June 2020.

### Study site

A total of 128 hospitals across HCMC were included in this study, of which 67 and 61 were public and private hospitals, respectively. The 67 public hospitals have been put under different administrations as follows: (i) 12 hospitals under the direct administration of the Vietnamese MOH, (ii) 32 hospitals under the administration of the HCMC Department of Health, and (iii) 23 district-level hospitals under the administration of the district-level People's Committee.

### Data collection and study instrument

Self-administered questionnaires were distributed to all CPs through an online tool named Zoho Survey. The questionnaire was initially pilot tested with 30 CPs and some amendments

were made to the wording and sequence of questions based on the pilot respondents' feedback. The questionnaire comprised three sections:

- Section one comprised demographic and work-related information of CPs: gender, age, marital status, academic degree, job position, income, number of duties that CPs handle from among 6 duties (professional pharmacy, storage and provision of drugs, pharmaceutical statistics, controlling the quality of drugs, managing the specialized activities of the hospital Pharmacy, and clinical pharmacy), time spent on tasks associated with clinical pharmacy, as well as with traditional pharmacy practice (5 out of the 6 duties mentioned above, apart from clinical pharmacy), and time spent on clinical ward rounds.

- Section two surveyed the clinical pharmacy tasks that CPs were currently performing. These tasks were extracted from Circular No. 31/2012/TT-BYT [4].

- Section three included the tool that was used to measure overall job satisfaction and the level of satisfaction with various factors that influence overall job satisfaction. Job satisfaction, which is the dependent variable, was measured by 3 items, whereas satisfaction with other factors (independent variables) was measured by 49 items. All 52 items were adapted from previously validated instruments in studies assessing job satisfaction among healthcare professionals and pharmacists, both nationally and internationally [13–17]. Participants were asked to rate their satisfaction on a 5-point Likert scale (1 = strongly dissatisfied, 5 = strongly satisfied).

   The data collection process went through three steps:

*Step 1*: Two separate survey links were emailed to all the Heads/Deputy Heads of pharmacy departments (HDPs) of hospitals in HCMC. Between these two links, one link was for HDPs *(Link A)* to report the number of CPs working in their hospitals; the objective here was to use this information to check whether all the CPs within each hospital had completed the survey or not. Meanwhile, the other link *(Link B)* contained the questionnaire used to survey CPs.

*Step 2*: HDPs from each hospital reported the number of CPs working there by completing the survey in Link A, and then forwarded Link B to CPs for their response.

*Step 3*: To ensure a good response rate, a reminder e-mail was sent to all HDPs 2 weeks later.

## Data analysis and statistical methods

Descriptive statistics (frequency, percentage, median, first quartile—third quartile [Q1-Q3], mean, and standard deviation [SD]) were used to describe the characteristics of the clinical pharmacy workforce, the clinical pharmacy tasks currently being performed by the CPs, and to summarize the main points related to CPs' satisfaction with their jobs and with different facets at work.

   Factor analysis was carried out to discover the underlying factors that could summarize the characteristics of the 49 items measuring the level of satisfaction toward different aspects; through this, only the item with a factor loading greater than 0.5 was retained [18]. After performing the factor analysis four times, nine items were eliminated and the remaining 40 items were summarized to seven factors: *(i) executives and internal regulation, (ii) income, (iii) trainings and promotion opportunities, (iv) job characteristics, (v) working conditions, (vi) inter & intra professional relationships* and *(vii) benefits (see* S1 Table*)*. Cronbach's alpha was calculated to ensure that each of these factors was reliable. Likewise, the internal consistency reliability of

the overall job satisfaction scale was also assessed using Cronbach's alpha. Each of these scales had to have a Cronbach's alpha value greater than 0.8 to meet the reliability requirement [18, 19]. The scores for overall job satisfaction and for the seven factors influencing job satisfaction of CPs were calculated by averaging the scores of items contained in each subscale. Next, multiple regression analysis was performed to evaluate the influential strengths of the seven factors on CPs' job satisfaction.

Additionally, CPs' satisfaction with the job and with the factors extracted from the factor analysis were grouped into a **"Not satisfied"** category (mean score ≤3.4, which consisted of three groups, namely "Strongly dissatisfied", "Dissatisfied", and "Neutral") and a **"Satisfied"** category (mean score >3.4, which consisted of two groups, namely "Satisfied" and "Strongly satisfied") [20]. Differences in overall job satisfaction by demographic and work-related characteristics were then calculated using the Chi-square/Fisher exact test. The statistical significance in the study was set at $p < 0.05$. All data were preprocessed using Microsoft Excel 2013 and analyzed using SPSS version 20.

### Ethics statement

This study was reviewed and approved by the ethics committees of Ho Chi Minh City Department of Health (No. 3082/SYT-NVD, dated June 1, 2020). Participants were fully informed of the purpose of the study, the procedures involved, and of the commitment to keep their information confidential. Their written consent was obtained via an informed consent form that was attached to the survey cover letter. By clicking on the "Start survey" button on the Zoho Survey tool, participants indicated their consent and willingness to participate.

## Results

Of the 128 HDPs who were approached to be a part of the study, 90 HDPs responded with the number of CPs and also forwarded the survey link to all CPs at their hospital. A total of 239 CPs reportedly worked in these 90 hospitals. However, by the end of the study, 197 out of the 239 CPs participated, thereby yielding a response rate of 82.4%. The information regarding the number of HDPs who responded and the number of questionnaires returned by CPs in each type of hospital is summarized in Fig 1.

### Characteristics of clinical pharmacists

The demographic and professional characteristics of the respondents are presented in Table 1. Regarding demographic information, most respondents were female (76.14%), below 30 years old (49.00%), and single (52.28%). Almost three-fourths of the CPs who participated in this study reported a bachelor's degree as their highest academic degree. Most of them (80%) had an income level lower than the average monthly income of the HCMC residents [21]. In terms of work-related characteristics, only about 30% of respondents were full-time CPs, while the remaining also had to handle other traditional pharmacy duties. The percentage of CPs participating in clinical ward rounds, along with the clinical care team, was relatively low (52.79%). Nevertheless, the time spent on the ward rounds was only a quarter of the total amount of time spent on clinical pharmacy activities. The average number of hours that CPs spent on clinical pharmacy activities per week was in fact, similar to the average number of hours spent on other conventional pharmacy duties (23.17 [SD: 14.11] hours/week and 22.92 [SD: 12.44] hours/week, respectively) (Fig 2).

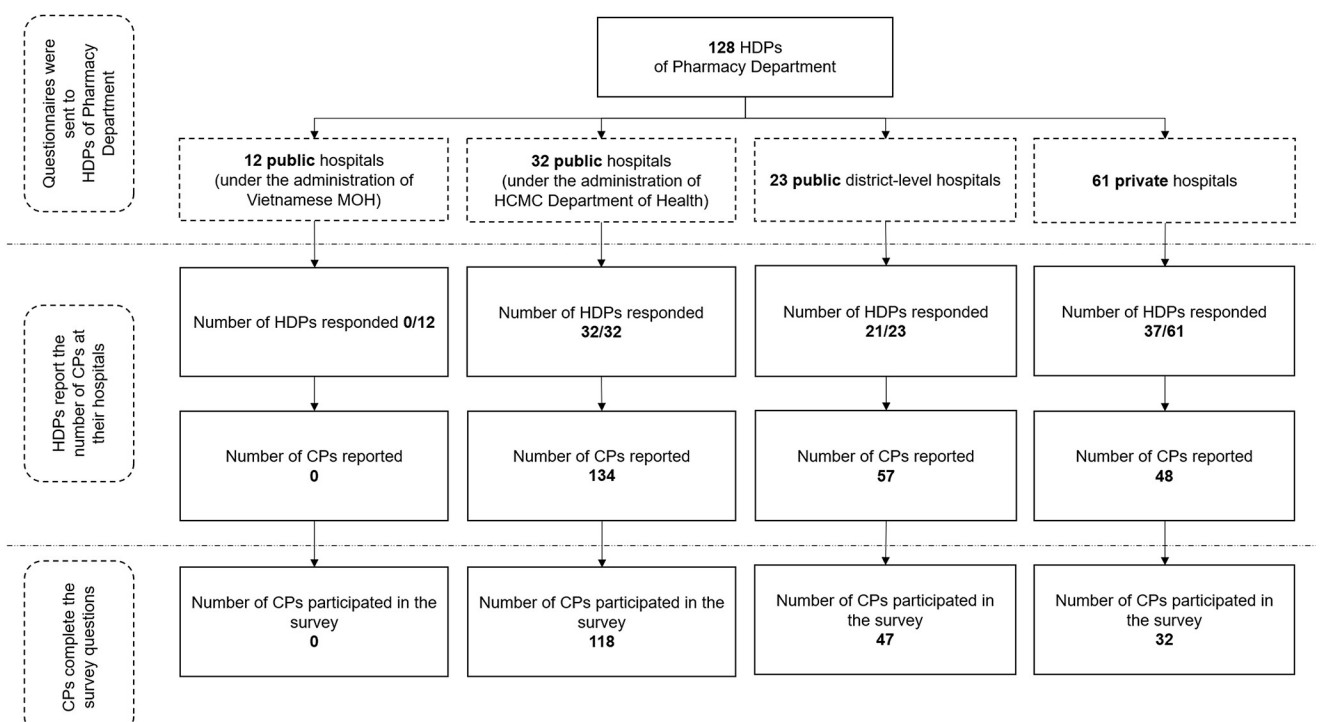

**Fig 1. Flow chart summarizing the data collection process.** Note: CPs: Clinical pharmacists, HDPs: Heads/Deputy heads, HCMC: Ho Chi Minh City, MOH: Ministry of Health.

## Clinical pharmacy activities

Table 2 illustrates the clinical pharmacy activities that were being implemented in hospitals in HCMC along with the number of CPs participating in each activity. These activities have been grouped into two: a group of general activities and a group of activities in clinical wards. In the former group, *"Provide drug information to patients and medical employees"* was the most common activity, with 88.3% CPs participating in it. Other activities that ranked lower in this group, in terms of the percentage of CPs performing them, were *"Participate in the analysis and evaluation of the drug use process"* (79.7%), *"Provide guidelines and supervision for the use of drugs in the hospital"* (78.7%), and *"Monitor and report the adverse reactions of drugs at the unit"* (76.1%). Furthermore, *"Participate in specialized consultation about drugs"* and *"Participate in developing and carrying out Therapeutic Drug Monitoring"*, were found to be the two least common activities that were performed by only 77 and 49 of the total CPs (below 40%), respectively. With regard to activities performed in clinical wards, it was indicated that three out of the four activities had a high proportion of CPs, performing them (above 70%), whereas the least performed activity in the clinical wards was *"Obtain information from the patient (through clinical record and patient interview)"* [n = 95 (48.2%)].

## Job satisfaction of clinical pharmacists

Overall, out of the 197 CPs who participated in this study, 74.1% reported being "satisfied" with their current job. Among the seven factors extracted from the factor analysis, the factor that the CPs were most satisfied with was *"Inter & Intra professional relationships"*, with a remarkably high percentage of CPs being satisfied with it (95.9). Meanwhile, only 59.9% of

**Table 1. Demographic and work-related characteristics (n = 197).**

| Demographic information | N (%) | Demographic information | N (%) |
|---|---|---|---|
| **Gender** | | **Income[a]** | |
| Male | 46 (23.35) | ≤456 USD/month | 157 (80) |
| Female | 150 (76.14) | >456 USD/month | 40 (20) |
| Other | 1 (0.51) | | |
| **Age** | | **Number of duties that CPs handle[b]** | |
| ≤30 | 97 (49) | Only 1 duty | 58 (29.44) |
| 31 to ≤40 | 71 (36) | 2 duties | 93 (47.21) |
| 41 to ≤50 | 21 (11) | ≥ 3 duties | 46 (23.35) |
| >50 | 8 (4) | | |
| **Marital status** | | **Ward round participation** | |
| Single | 103 (52.28) | Yes | 104 (52.79) |
| Married | 93 (47.21) | No | 93 (47.21) |
| Others | 1 (0.51) | | |
| **Academic degree[c]** | | **Job position** | |
| B.Pharm | 144 (73.10) | Head of Pharm.Dept | 11 (5.58) |
| M.Pharm | 27 (13.71) | Deputy Head of Pharm.Dept | 14 (7.11) |
| Ph.D.Pharm | 1 (0.51) | Official employee | 118 (59.90) |
| F.D.S.Pharm | 20 (10.15) | Short-term employee | 46 (23.35) |
| S.D.S. Pharm | 5 (2.54) | On-probation employee | 8 (4.06) |

[a]Average income of Ho Chi Minh City residents.

[b]Only 1 duty = Only clinical pharmacy duty; 2 duties = Clinical pharmacy + 1 traditional pharmacy duty; ≥ 3 duties = Clinical pharmacy + ≥ 2 traditional pharmacy duties.

[c]B. Pharm: Bachelor of Pharmacy; F.D.S. Pharm: First-level Diploma of Specialization in Pharmacy; S.D.S. Pharm: Second-level Diploma of Specialization in Pharmacy; M. Pharm: Master of Pharmacy; Ph.D. Pharm: Doctor of Philosophy in Pharmacy.

CPs reported being satisfied with the factor *"Income"*, which was also the factor that possessed the lowest level of satisfaction. The remaining five factors led to a high level of satisfaction among CPs (more than 75% reported satisfaction) (Table 3). The items from each factor that CPs reported to be least satisfied with are presented in Table 4. The complete results regarding CPs' satisfaction with each item from each factor are presented in S1 Table.

As shown in Table 3, two out of the seven factors did not significantly predict job satisfaction, namely *"Trainings and promotion opportunitie"* and *"Benefits"*. The remaining five factors including *"Executives and internal regulation"*, *"Income"*, *"Job characteristics"*, *"Working conditions"* and *"Inter & Intra professional relationships"* positively impacted current job satisfaction of the CPs; specifically, *"Executives and internal regulation"* was found to be the most important predictor of job satisfaction, and even an increase of 1 unit in the satisfaction score for this factor increased the overall job satisfaction average score by 0.353 units. Meanwhile, the other four factors contributed a relatively comparable level of increase (ranging from 0.11 to 0.176 units) in the overall job satisfaction score when the satisfaction scores of each of these factors increased by 1 unit.

The association between demographic and work-related characteristics and job satisfaction is depicted in Table 5. The only characteristic noted to have a statistically significant association with overall job satisfaction was "Ward round participation" (p = 0.004); that is, a higher percentage of CPs (82.7%) who performed activities in clinical wards reported feeling satisfied with their job as compared to those CPs who did not participate in this activity (64.5%).

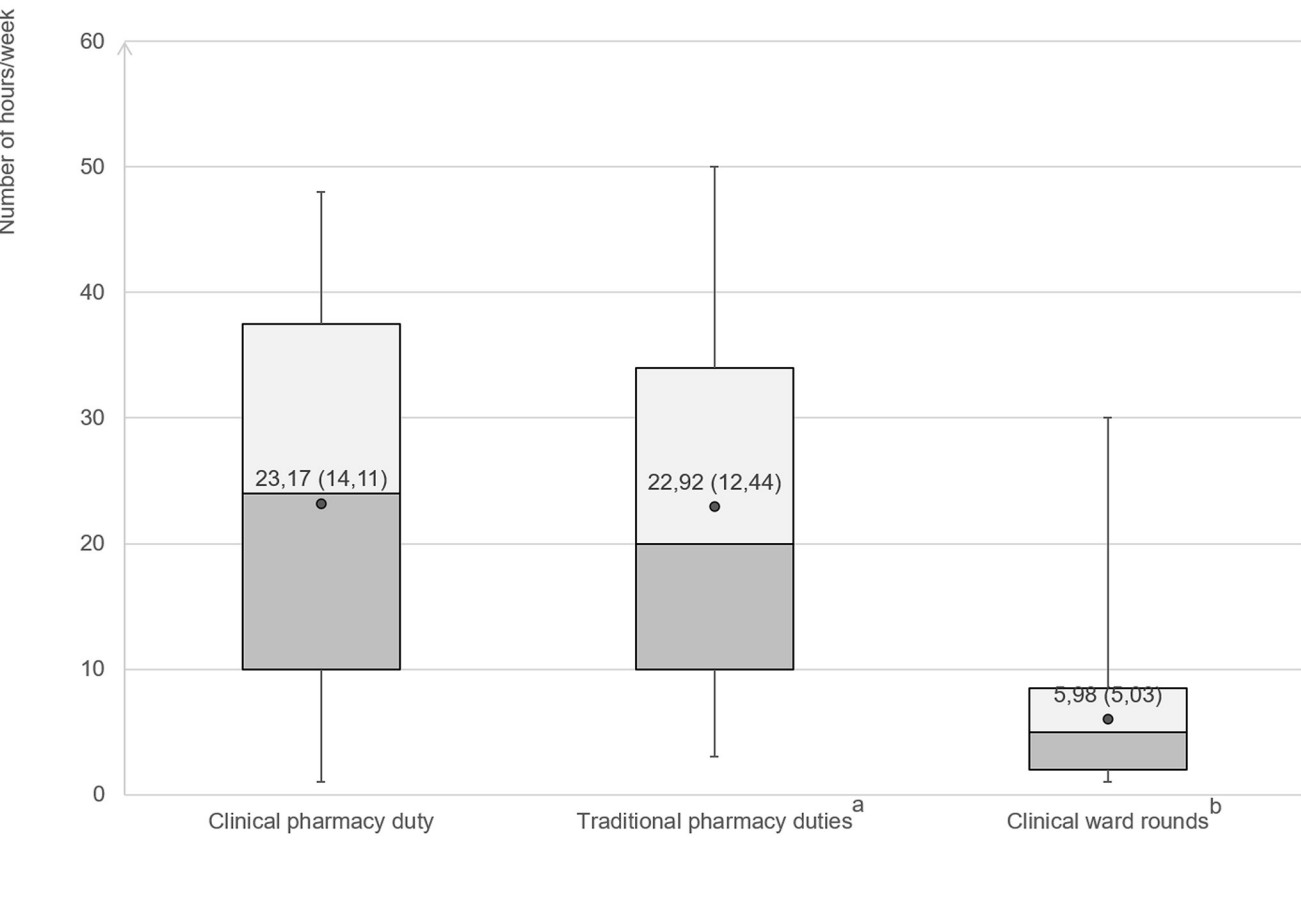

**Fig 2. Mean time that Clinical Pharmacists (CPs) spent on clinical pharmacy duty, traditional pharmacy duties, and clinical ward rounds [mean (standard deviation)].** [a]CPs who handled only clinical pharmacy duty were not included, [b]CPs who did not participate in ward rounds were not included.

## Discussion

### Clinical pharmacy workforce characteristics

Until June 2020, the total number of CPs working in 90 hospitals was 239, which corresponds to a ratio of 2.7 CPs per hospital. In general, the average number of CPs across HCMC reported in this study (2.7 CPs/hospital) was not significantly different from the figures reported for Hanoi in 2010 and 2015, which were 2.6 and 2.2 CPs/hospital, respectively [22]. Additionally, this ratio was even lower than the standard in other Asian countries, namely China and India, where each hospital must have at least 3 CPs [23, 24]. Meanwhile, Bond et al., in a study conducted in 2004, proved that, on average, a hospital needed approximately 11 CPs to ensure effective implementation of the five core clinical pharmacy tasks [25]. Furthermore, almost three-fourths of the 197 CPs who participated in this study reported a B. Pharm (Bachelor of Pharmacy) degree as being their highest degree (73.1%), while those who possessed a second-level Diploma of Specialization in Pharmacy and a Doctor of Philosophy degree in Pharmacy (two highest academic levels in pharmacy education in Vietnam) comprised an extremely low percentage within the study sample, at 2.54% and 0.51%, respectively. These results suggest that hospitals in HCMC face a shortage of CPs, especially those CPs who have a

**Table 2. Number of clinical pharmacists participating in activities defined by regulation 31 (n = 197).**

| Groups of task | N (%) |
|---|---|
| **General activities** | |
| Provide drug information to patients and medical employees | 174 (88.3) |
| Participate in the analysis and evaluation of the drug use process | 157 (79.7) |
| Provide guidelines and supervision for the use of drugs in the hospital | 155 (78.7) |
| Monitor and report the adverse reactions of drugs at the unit | 150 (76.1) |
| Training in clinical pharmacy | 129 (65.5) |
| Participate in the periodic clinical case study in the hospital | 122 (61.9) |
| Provide monthly, quarterly, annual and surprise reports about the drug use activities | 112 (56.9) |
| Strictly supervise compliance with the drug use process | 101 (51.3) |
| Participate in developing specialized processes related to the use of drugs | 96 (48.7) |
| Participate in scientific research work and activities | 95 (48.2) |
| Participate in consultation during the process of developing the unit's list of drugs | 85 (43.1) |
| Participate in developing the process of supervising the use of listed drugs | 84 (42.6) |
| Participate in specialized consultation about drugs | 77 (39.1) |
| Participate in developing and carrying out Therapeutic Drug Monitoring | 49 (24.9) |
| **Activities in the clinical wards** | |
| Examine the drugs prescribed to the patients | 164 (83.2) |
| Provide drug instructions to nurses | 145 (73.6) |
| Cooperate with physicians to provide counseling for patients | 141 (71.6) |
| Obtain information from the patient (through clinical record and patient interview) | 95 (48.2) |

higher level of qualification. This conclusion is similar to the one drawn in another study in 2019 [5]. Therefore, hospitals need to urgently speed up the process of recruiting CPs, while simultaneously supporting and encouraging their clinical pharmacy staff to pursue postgraduate studies to improve both their professional expertise and experience. Meanwhile, many countries all over the world, such as the United States, Canada, Korea, Japan, Pakistan, Thailand, etc. have been gradually eliminating the B. Pharm degree and making it mandatory for a pharmacist to possess a Pharm. D (Doctor of Pharmacy) degree to participate in clinical pharmacy activities (the Pharm. D is a doctorate degree similar to Ph. D. Pharm, except that it is a professional or a more clinically oriented degree, whereas a Ph. D. Pharm is a research

**Table 3. Reliability coefficients, level of satisfaction and regression coefficients of various factors related to current job satisfaction (n = 197).**

| Factor | Cronbach's Alpha | Mean (SD) | Median (Q1-Q3) | No. of CPs satisfied[a] [n (%)] | Regression coefficient (b)[b] | P value |
|---|---|---|---|---|---|---|
| **Executives and internal regulation** | 0.945 | 3.98 (0.57) | 4.00 (3.78–4.22) | 173 (87.8) | 0.353 | <0.001 |
| **Income** | 0.940 | 3.54 (0.75) | 3.80 (3.00–4.00) | 118 (59.9) | 0.110 | 0.01 |
| **Trainings and promotion opportunities** | 0.894 | 3.80 (0.62) | 4.00 (3.50–4.00) | 155 (78.7) | * | * |
| **Job characteristics** | 0.920 | 3.88 (0.55) | 4.00 (3.60–4.00) | 162 (82.2) | 0.168 | 0.008 |
| **Working conditions** | 0.883 | 3.88 (0.54) | 4.00 (3.67–4.00) | 168 (85.2) | 0.143 | 0.02 |
| **Inter & Intra professional relationships** | 0.871 | 4.07 (0.44) | 4.00 (3.83–4.17) | 189 (95.9) | 0.176 | 0.02 |
| **Benefits** | 0.883 | 3.99 (0.59) | 4.00 (4.00–4.00) | 176 (89.3) | * | * |
| **Overall job satisfaction** | **0.821** | **3.81 (0.55)** | **4.00 (3.33–4.00)** | **146 (74.1)** | - | - |

*Not statistically significant at the time of running the first regression; thus, these factors were eliminated.

[a]Number of CPs with a mean score of satisfaction >3.40 ("Satisfied" and "Strongly satisfied").

[b]Constant = 0.097; $R^2$ = 66.5%.

**Table 4. The least enjoyable item from each factor (n = 197).**

| Factor | Least enjoyable item | No. of CPs satisfied[a] [n (%)] |
|---|---|---|
| Executives and internal regulation | Hospital executives encourage and praise you when you perform excellently at work | 146 (74.1) |
| Income | Your salary matches up to your competence and contribution | 94 (74.7) |
| Trainings and promotion opportunities | Hospital's promotion policy is transparent and reasonable | 126 (64) |
| Job characteristics | Your current job is interesting | 140 (71.1) |
| Working conditions | Your workplace is equipped with sufficient clinical pharmacy-related materials | 140 (71.1) |
| Inter & Intra professional relationships | You are able to work well in collaboration with physicians/nurses | 165 (83.8) |
| Benefits | You are offered vacations regularly | 169 (85.8) |

[a]Number of CPs who rated these items 4 or 5 on the Likert-5 scale ("Satisfied" and "Strongly satisfied").

graduate degree) [26]. Further, as per a publication on pharmacy education in Vietnam, the B. Pharm degree in general is more product-oriented and mainly focuses on laboratory-based courses, while clinical training and practical experience are not given adequate attention. As a result, pharmacy graduates in Vietnam tend to excel more in the pharmaceutical industry, such as drug manufacturing, research, quality assurance and control, and drug discovery, but when it comes to clinical pharmacy practice, most of them are not equipped with sufficient expertise and hands-on experience [27]. This weakness, however, can be entirely surmounted with introducing a Pharm. D curriculum. Therefore, administrative agencies should consider creating a clear pathway for the transformation of CPs' academic degrees from B. Pharm to

**Table 5. Job satisfaction in relation to demographic and work-related characteristics (n = 197).**

| Characteristics | Satisfied (n) | Not satisfied (n) | P value | Characteristics | Satisfied (n) | Not satisfied (n) | P value |
|---|---|---|---|---|---|---|---|
| **Gender**[a] | | | | **Academic degree** | | | |
| Male | 39 | 7 | 0.067? | B.Pharm | 106 | 38 | 0.791 |
| Female | 107 | 43 | | Postgraduate | 40 | 13 | |
| **Age** | | | | **Income** | | | |
| ≤30 | 72 | 25 | 0.971 | ≤456 USD/month | 117 | 40 | 0.794 |
| >30 | 74 | 26 | | >456 USD/month | 29 | 11 | |
| **Marital status**[a] | | | | **Ward round participation** | | | |
| Single | 75 | 28 | 0.696 | Yes | 86 | 18 | 0.004 |
| Married | 70 | 23 | | No | 60 | 33 | |
| **Job position** | | | | **Number of duties that CPs handle**[d] | | | |
| HDPs[b] | 22 | 3 | 0.219 | Only 1 duty | 47 | 11 | 0.356 |
| Official employee | 84 | 34 | | 2 duties | 66 | 27 | |
| Others[c] | 40 | 14 | | ≥ 3 duties | 33 | 13 | |

[a]Sum does not equal 197 as some groups were excluded to satisfy the Chi-square/Fisher test condition.

[b]Heads/Deputy heads of Pharmacy Department.

[c]Short-term employee, on-probation employee.

[d]Only 1 duty = Only clinical pharmacy duty; 2 duties = Clinical pharmacy + 1 traditional pharmacy duty; ≥ 3 duties = Clinical pharmacy + ≥ 2 traditional pharmacy duties.

Pharm. D, in terms of both clinical pharmacy education and practice, in order to achieve the long-term objectives of raising the quality of the clinical pharmacy workforce and catching up with the rest of the world.

## Status of clinical pharmacy activities

Only 30% of the participants in this study were full-time CPs, while the remaining had to handle at least 2 duties simultaneously. Furthermore, the time that CPs spent on activities that do not relate to clinical pharmacy nearly equaled the time spent on clinical pharmacy activities. This might result in clinical pharmacy activities being negatively impacted since CPs are unable to fully concentrate on their specialized tasks. In addition, the proportion of CPs participating in clinical ward rounds with physicians and nurses was relatively low (53%), and the time spent on this activity was only 6 hours/week. In two studies conducted in 2014 [28] and 2015 [22] in Hanoi, the proportions of CPs participating in clinical ward rounds were 52% (with 10 hours spent on this per week) and 44% (with 7.2 hours spent per week), respectively. Our results reflect a similarity in ward round participation between CPs in Hanoi and in HCMC, the first two municipalities to take the lead in including clinical pharmacy in the healthcare practice area.

On the other hand, the time spent on ward rounds varied remarkably among CPs (ranging from 1 hour/week to 30 hours/week), with about 25% of CPs spending at least 8.5 hours/week on this activity. In conclusion, although ward round participation was distributed disproportionately among hospitals in HCMC, this activity has also gradually been receiving the most attention, which brings clinical pharmacy nearer to its objective of being "patient-centered".

A high rate of participation was observed among the CPs for most of the activities (14 general activities and 4 clinical ward activities) stipulated in Circular No. 31/2012/TT-BYT, even though there was an uneven distribution of CPs across the activities. Of the 14 general activities, the most commonly reported activities (representing 75% to 90% of the total activities; Table 2) were also those that previous studies found were being implemented in majority of hospitals (≥90%) in Vietnam [22, 28]. Additionally, three of the four clinical ward activities had a high percentage of CPs participating in them (70–80%); the percentage for "Exploit information from the patient (through clinical record and patient interview)," however, was found to be relatively low (48.2%), despite this being a core activity and a significantly important skill for CPs [29]. Moreover, this particular activity has been proven to be beneficial for healthcare practice in many ways. For instance, medication-related problems can be more effectively detected in groups of patients that are directly interviewed by CPs; similarly, some special issues can only be determined by combining information from both patient-records and patient interviews, which further gives rise to higher treatment effectiveness [30–32]. Therefore, hospitals should regularly conduct training and continuing medical education programs, as well as build elaborate guidelines that can assist CPs in enhancing this vital skill. This could become a strong cornerstone for the gradual development and strengthening of clinical pharmacy services in HCMC in the future.

## Job satisfaction of clinical pharmacists

Results about job satisfaction indicated that "Executives and internal regulation" was the factor that led to the largest increase in overall job satisfaction (0.353 units in average score), while "Income," "Job characteristics," "Working conditions," and "Inter & Intra professional relationships" contributed to a relatively comparable level of increase (ranging from 0.11 to 0.176 units) in the overall satisfaction level.

**Executives and internal regulation.**   In hospitals, executives tend to be increasingly focused on the role of CPs, and the internal regulations aim to create a suitable, professional, and ideal working environment that improves the productivity and working efficiency of CPs. The high level of satisfaction among CPs shows that the leadership and the regulatory policies in the hospital are reasonable and effective (*individual component results are detailed in* S1 Table). As a result, not only did CPs report being highly satisfied with this factor but it also positively affected their overall job satisfaction. This primarily stems from the development orientation of the hospitals. Specifically, nowadays, due to the autonomous mechanism, hospitals tend to focus on improving the quality of medical examinations and treatments in order to attract more patients. This further requires them to also comply with mandatory regulations of the government (Decree 131/2020/ND-CP) and, therefore, have appropriate remuneration policies and internal regulations to attract and retain CPs. Since there are several challenges with recruitment due to the influence of pay or the procedures and regulations in recruitment, the presence of appropriate internal regulations and the direction provided by the leader greatly influence the CPs' decision to pursue long-term clinical pharmacy work seriously [5]. However, on the other hand, CPs also tend to be younger in age and they want to be attached to and develop in the clinical field; thus, their satisfaction level is likely to be affected by their love for the work as well. Considering this, satisfaction with leadership and with internal regulations can be seen as being temporary. Therefore, it is necessary to conduct further studies to consider and analyze the causes that can influence this result; this will help in gaining more awareness about the application of the current research results.

The relationship between leadership style and job satisfaction of health workers has been demonstrated in studies around the world [33–35]. In Vietnam, when the role of an executive is always at the center, the executives and the internal regulations established by them exert great influence on employees' trust and their level of job satisfaction. This is the basis for leaders to consider having a long-term leadership vision, as well as to develop and expand suitable internal regulations. Hence, the satisfaction level of CPs at the hospital were found to have been increasing.

**Inter & intra professional relationships.**   Among the five factors that significantly impacted overall job satisfaction, *"Inter & Intra professional relationships"* was noted to be the factor leading to the highest level of satisfaction among CPs, with 95.9% CPs being "satisfied" with this. As depicted by the results, the relationship between CPs and their colleagues was considered to be extremely ideal and a positive sign; a working environment that gives rise to jealousy and conflict among co-workers might deter individual development, as well as overall job productivity. However, within this factor, "working in collaboration with physicians and nurses" resulted in the lowest level of satisfaction. Therefore, despite high satisfaction among CPs regarding their relationships at the workplace, it is vital to further focus on improving their relationship with physicians/nurses in order to ensure smooth and effective collaboration between these core members of the clinical care team. A few ways in which this goal can possibly be attained are: (i) providing more opportunities for CPs to interact with physicians and nurses, (ii) developing elaborate clinical pharmacy guidelines that clearly define the specialized role of each member (physician, nurse, and CP) within the clinical care team, and (iii) raise awareness among physicians and nurses about the role of CPs. Such simple measures will allow CPs to confidently express their opinions and freely discuss medication-related aspects with other healthcare professionals, thereby contributing to an increase in the quality of clinical pharmacy services and health care for patients.

**Job characteristics.**   Within this factor, the lowest percentage of CPs reported satisfaction with the item *"Your current job is interesting"* (71.1%). This might have resulted from the fact that hospitals in HCMC have been lacking an adequate and well-qualified clinical pharmacy

workforce, and CPs, are required to handle manifold duties simultaneously, as previously discussed. Furthermore, CPs are also expected to devote a significant amount of their time participating in many traditional hospital pharmacy activities, which are deemed repetitive and boring; meanwhile, the time spent on novel and challenging clinical pharmacy tasks remains limited. Consequently, in order to enhance CPs' satisfaction with this factor, it is necessary to adopt policies relating to CP recruitment and work assignment which ensure that there are enough pharmacists available to perform conventional hospital pharmacy tasks, there is high level of specialization involved in clinical pharmacy activities, and CPs are allowed sufficient time to concentrate on cultivating their professional skills.

**Working conditions.** As part of this factor, the sub-item that CPs reported being least satisfied with was "Your workplace is equipped with sufficient clinical pharmacy-related materials," with 71.1% of CPs feeling "satisfied". Given that clinical pharmacy is an evidence-based practice, every decision made by CPs must be supported by transparent and updated scientific evidence. Therefore, the CP's workplace should be equipped with more diversified materials related to pharmacology, clinical pharmacy, drug interaction, and antibiotic use, and CPs should be provided with the means to access online resources that assist them in effortlessly answering various practice-related questions.

**Income.** *"Income"* resulted in the lowest level of satisfaction, with only 59.9% CPs being satisfied with this factor. Although the influence of the four factors, namely *"Inter & Intra professional relationships"*, *"Job characteristics"*, *"Working conditions"* and *"Income", was* nearly equal, the CPs still reported much lower satisfaction with "Income" than with the other factors. Low income satisfaction has also been observed among other healthcare professionals in many studies conducted in other countries [15, 36–38], as well as in various provinces in Vietnam [17, 39, 40]. The biggest cause for dissatisfaction among CPs, with respect to income, was that the salary did not match their competence and contribution to the job. While the salary of healthcare professionals in Vietnam is often fixed based on Decree No. 38/2019/ND-CP and Decree No. 204/2004/ND-CP, consequently making it difficult to make any further adjustments, the absence of a reasonable income increases the risk of CPs resigning from their jobs to pursue other jobs that may offer a more reasonable income [41, 42]. This can have serious repercussions for hospitals since recruiting and training new CPs can be a costly and time-consuming task, which, in turn, hampers the development of clinical pharmacy. In view of this, in order to raise CPs' satisfaction with their income and to subsequently retain them in their current job for longer, other sources of income, such as allowances, bonuses, and rewards should be offered.

**Overall job satisfaction and its association with demographic/work-related characteristics.** Overall, 74.1% of the CPs included in this study felt satisfied with their current job. This figure was lower than that noted in Japan (95%) and China (around 90%), but higher than that noted in Malaysia (52%), Saudi Arabia (47%), Ethiopia (32.7%), and in other countries with a more developed clinical pharmacy profession such as the United States (67.3%) and Ireland (60%) [36–38, 43–46]. Within Vietnam, there was a similarity noticed in the level of job satisfaction reported by CPs in this study and that reported by all types of healthcare professionals in Vinh Phuc province and Kon Tum province, where satisfied employees represented 71.1% and 72% of the total employees surveyed, respectively [39, 40]. Meanwhile, job satisfaction among hospital pharmacists in district-level hospitals in Vietnam in 2015 was significantly low, with the proportion of satisfied CPs being 24.9% [17]; compared to this, CPs in hospitals in HCMC reported higher satisfaction with their current jobs. Nevertheless, this high level of satisfaction is less likely to remain unchanged if hospitals' policies that are meant to benefit CPs also remain constant. This prediction stems from the fact that clinical pharmacy is relatively new in Vietnam and HCMC, and thus, difficulties and deterrents during practice are

inevitable. As pioneers in this novel pharmacy practice, CPs have been more conscious of this issue than anybody else and have been more willing to adapt to difficulties at work, including a lower pay. In other words, until now, it has been easier for CPs to feel satisfied with their job. However, it is safe to assume that CPs' requirements and expectations regarding their jobs might progressively increase with the growth of clinical pharmacies in the near future. Furthermore, while pharmacy graduates have a variety of professional choices that are associated with a high income, CPs are often required to handle a heavy workload with high pressures and responsibilities [5]. All these reasons might contribute to a greater probability of CPs quitting their jobs. Therefore, CPs' job satisfaction should be one of the top concerns that are persistently attended to by the hospitals' pharmacy department and board of director to maintain long-term organizational and job commitment among CPs.

Regarding job satisfaction and demographic/work-related characteristics, among CPs who were satisfied with their current job, a statistically significant difference was observed only between these two groups: those who participated in clinical ward rounds (82.7%) versus those who did not (64.5%); this indicated the importance of ward round participation in job satisfaction of CPs. Hence, hospitals should strongly encourage CPs to participate in ward rounds, which would enable them to apply their acquired knowledge and skills into practice regularly, as well as to increase their level of interest in the current job. Additionally, the presence of CPs in the clinical care team would certainly allow them to meaningfully contribute to treatment effectiveness.

## Study limitations

This study could not obtain information from all CPs working in HCMC because it was conducted during the COVID-19 pandemic and some CPs, therefore, did not find the time to complete the survey. Moreover, CPs' opinions and proposals on how to increase the effectiveness of clinical pharmacy services, as well as their job satisfaction, were not surveyed, which acts as a deterrent for the boards of directors at hospitals and at the HCMC Department of Health when formulating the most optimal and practical policies.

## Conclusion

In conclusion, clinical pharmacy in HCMC started with, and so far has been, yielding favorable outcomes. Nevertheless, this novel field is still coping with numerous limitations and barriers, including CP shortage, low income for CPs, lack of synchronization in implementing clinical pharmacy activities among hospitals, lack of specialization at work, etc. Additionally, with regard to aspects related to CPs' job satisfaction, two factors that currently need to be prioritized through hospitals' policies are CPs' income and ward round participation.

## Supporting information

**S1 Table. Factor analysis result and clinical pharmacists' satisfaction on each item (n = 197).**
(DOCX)

**S1 File.**
(DOCX)

## Acknowledgments

The authors would like to thank the HDPs as well as CPs in hospitals at HCMC for their participation in this study. We also would like to thank Editage (www.editage.com) for English language editing.

## Author Contributions

**Investigation:** Thuy-Tram Nguyen-Ngoc, Minh-Thu Do-Tran, Dung Van Do.

**Methodology:** Thuy-Tram Nguyen-Ngoc.

**Project administration:** Hai-Yen Nguyen-Thi, Luyen Dinh Pham, Nguyen Dang Tu Le.

**Resources:** Dung Van Do.

**Software:** Thuy-Tram Nguyen-Ngoc.

**Supervision:** Hai-Yen Nguyen-Thi, Luyen Dinh Pham, Nguyen Dang Tu Le.

**Validation:** Thuy-Tram Nguyen-Ngoc, Nguyen Dang Tu Le.

**Visualization:** Thuy-Tram Nguyen-Ngoc.

**Writing – original draft:** Thuy-Tram Nguyen-Ngoc, Minh-Thu Do-Tran.

**Writing – review & editing:** Nguyen Dang Tu Le.

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
