## [Decision Letter · Decision Letter 0]

19 Nov 2020

PONE-D-20-31080

Job satisfaction of clinical pharmacists and clinical pharmacy activities implemented at Ho Chi Minh City, Vietnam

PLOS ONE

Dear Dr. Nguyen-Thi,

Thank you for submitting your manuscript to PLOS ONE. After careful consideration, we feel that it has merit but does not fully meet PLOS ONE’s publication criteria as it currently stands. Therefore, we invite you to submit a revised version of the manuscript that addresses the points raised during the review process.

As noted by the reviewers, there are a number of areas in the manuscript that require clarification.  Please address each of these issues as noted by the reviewers.

We look forward to receiving your revised manuscript.

Kind regards,

John Rovers, PharmD, MIPH

Academic Editor

PLOS ONE

Journal Requirements:

3. In the ethics statement in the Methods section and online submission information, please specify the type of informed consent that was obtained from the participants (for instance, written or verbal, and if verbal, how it was documented and witnessed).

4. As part of your revision, please complete and submit a copy of the COREQ Guidelines checklist, a document that aims to improve reporting of qualitative studies for purposes of post-publication data analysis and reproducibility: https://www.equator-network.org/reporting-guidelines/coreq/. Please include your completed checklist as a Supporting Information file. Note that if your paper is accepted for publication, this checklist will be published as part of your article.

5. Please include additional information regarding the survey or questionnaire used in the study and ensure that you have provided sufficient details that others could replicate the analyses. For instance, if you developed a questionnaire as part of this study and it is not under a copyright more restrictive than CC-BY, please include a copy, in both the original language and English, as Supporting Information, or include a citation if it has been published previously.

6. In the Methods, please discuss whether and how the questionnaire was validated and/or pre-tested. If these did not occur, please provide the rationale for not doing so.

7. We note that you have indicated that data from this study are available upon request. PLOS only allows data to be available upon request if there are legal or ethical restrictions on sharing data publicly. For more information on unacceptable data access restrictions, please see http://journals.plos.org/plosone/s/data-availability#loc-unacceptable-data-access-restrictions.

8. PLOS requires an ORCID iD for the corresponding author in Editorial Manager on papers submitted after December 6th, 2016. Please ensure that you have an ORCID iD and that it is validated in Editorial Manager. To do this, go to ‘Update my Information’ (in the upper left-hand corner of the main menu), and click on the Fetch/Validate link next to the ORCID field. This will take you to the ORCID site and allow you to create a new iD or authenticate a pre-existing iD in Editorial Manager. Please see the following video for instructions on linking an ORCID iD to your Editorial Manager account: https://www.youtube.com/watch?v=_xcclfuvtxQ

Reviewers' comments:

Reviewer's Responses to Questions

**Comments to the Author**

1. Is the manuscript technically sound, and do the data support the conclusions?

Reviewer #1: Yes

Reviewer #2: Yes

2. Has the statistical analysis been performed appropriately and rigorously? 

Reviewer #1: Yes

Reviewer #2: Yes

3. Have the authors made all data underlying the findings in their manuscript fully available?

Reviewer #1: Yes

Reviewer #2: No

4. Is the manuscript presented in an intelligible fashion and written in standard English?

Reviewer #1: Yes

Reviewer #2: Yes

5. Review Comments to the Author

Reviewer #1: Comments to authors

This an interesting article in the field of pharmacy practice. Some observations and limitations have to be considered into account before publication.

In abstract: more details about the method of data collection is better to be given.

In the introduction: the objectives of the study are not clear and the term “To summarize” is not acceptable as clear objective and cannot express the effort done by the researchers. I suggest to change it “To describe the status of clinical pharmacy profession implementation and analyze the key factors affecting job satisfaction of CPs at HCMC”

In the method; in line 102 written that data collection process went through four steps, while details of steps followed and figure 1 diagram shows 3 steps only. In factor analysis more explanation was needed regarding factor loading analysis and to justify why retained factor loading greater than 0.5 and eliminated less than that and on what base done that.

In Results: in table (2) the 7th general task the word “surprise” has no meaning and I think you may be meant to say “supervise”. Term “Colloquies” is not appropriate as a factor for job satisfaction and I suggest to change it “ Inter & Intra professional relationship”

Discussion: No comments.

General: English language needs to be revised.

Reviewer #2: PONE-D-20-31080: Job satisfaction of clinical pharmacists and clinical pharmacy activities implemented at Ho Chi Minh City, Vietnam

Thank you for the opportunity to review this research. I believe this is a strong study describing the factors that are impacting the developing professional practice of clinical pharmacists in Vietnam. This work makes a contribution to understand what may be needed to foster the professionalization and clinical development of professionals in this field within the constraints of their current work environment. Further, the survey research approach used a variety of statistical analysis to appropriately assess the cross-sectional data from a sample of clinical pharmacists. The study discusses the implications of the findings in context of clinical pharmacy within other parts of the country and other parts of the world, and provides some ideas for what this means for policy and administrative changes at the local and national levels.

The study biggest strength are the methodological approach and the application of a variety of statistical analysis that very comprehensively breaks down the nuances in the data to reveal the feelings, perceptions, and current work tasks of clinical pharmacists in HCMC.

The biggest weakness of the study is in the description of the assessed outcomes, including the categories of work. Is was difficult to follow the ideas about tasks and measured items, and for some, the way they were grouped together. I will elaborate more on this below, relevance to each specific section of the manuscript.

ABSTRACT

LINE 26: In the background in the abstract is states “numerous difficulties” but specific difficulties are never elaborated in the main manuscript, so I recommend to change the sentence to be more neutral. The study is about both positive and negative aspects that impact work, therefore I suggest “numerous factors” as a preferable framing in the background.

LINE 42-43: States “favorable outcomes” and “limitations and barriers” I recommend to change this sentence that is more specific, with whichever the authors think are most noteworthy from the main conclusions.

INTRODUCTION

LINE 50: It would be helpful for a reader unfamiliar with Vietnam policy (and also unable to locate Vietnam legal records) that the authors consider to add a brief description about the first policy including some key roles/tasks for CPs that are in this policy. In addition, as a reader I am interested to know what was the main rationale or cause for this policy to be created. For example, was there a growing rate of medication errors that caused this? Was it because of request from the hospitals? Was the policy a result of the work of professional advocates?

LINE 54: “barriers” and “limitations” that are described as being a problem for CPs. I understand that the authors refer to previous reports and previous research. However, given that these ideas are foundational to the present investigation, a brief explanation as to the nature of the problems would be helpful. This will also help assessment the alignment between the foundation of the study, the chosen measures and the conclusions. A simple sentence that states, “For example…..” would likely help orient the reader.

LINE 66-68 This section is talking about the important of assessing job satisfaction. To help make this claim more credible, it is recommend to add a citation of previous research that makes the link between job satisfaction and quality of work or job performance.

METHODS

LINE 75: The design mentions qualitative, but I don’t recall the results of this part in the results section. Can the authors clarify what aspects were evaluated for each of the qualitative and quantitative sections from the study?

LINE 88-90: It is not clear what the tasks are that are called “tasks concurrently handled” compared to “clinical pharmacy tasks” and “other traditional pharmacy practice.” Clinical pharmacy tasks are specific to the new law, but it is recommended that more clear examples to help distinguish what it means by a “task concurrently handled” means. For example, is this non-pharmacy tasks like cleaning or something else?

Line 91: It is unclear why “time spend on ward round participation” is separate from “clinical pharmacy.” It seems these are both in the same category of ideas that a CP should do. Explain why this task is separated out from the rest.

RESULTS

Line 159: Similar to previous comments, it is unclear what is meant by “concurrently handle various tasks” Some examples in the methods or results section would help the reader understand what these mean. Does this mean they have 2 different jobs?

Figure 2 needs more information in the figure so that the figure can stand-alone for interpretation. As to previous comments, it is not clear how on-ward participation is different from clinical pharmacy. Are these in the same category? Also, does the middle bar (about “other” duties include “tasks concurrently handled” as well as traditional pharmacy services? I think if these categories were better described in the methods, then in the results would be more clear. Also the figure need to give a title.

LINE 170-181: I am not sure what on ward rounds is not included in this assessment of clinical pharmacy tasks. However, the clinical pharmacy tasks appear to be slip between on and off the ward duties. I think this confusion can be clarified with the definitions and rationale for these categories within the methods section. Then, the results will be more understandable.

DISCUSSION

LINE 247: This sentence and ideas about how the PhD Pharm degree is different needs to be re-written for clarity. It is unclear how this is different or similar to the PharmD degree. “…Focus more on practice aspect” needs to be clarified.

LINE 249: A stronger case will need to be made of the rationale of the need to increase credentialing for CPs. If they are currently fulfilling the duties as required, what evidence can be provided to correlate increased credential to increase workforce quality?

LINE 255: The authors compare full-time CPs with others who have the handle “at least 2 tasks” – this makes me believe that this mean like jobs different from pharmacy. However, see earlier comments about clarifying what this work category means.

LINE: 299: The idea that executives, and internal regulations impact job satisfaction is a very interesting finding. Therefore, I believe the authors should discuss potential ways that these policies do to impact CPs. Why do the authors believe this was the biggest factor?

General Comments:

Recommend remove the use of the term “prove” as this is too definitive for being used to refer to ongoing study or only one or a few research studies.

6. PLOS authors have the option to publish the peer review history of their article (what does this mean?). If published, this will include your full peer review and any attached files.

Reviewer #1: **Yes: **Dr. Salah I Kheder - Associate professor of Pharmacology

Reviewer #2: No

---

## [Author Response · Author response to Decision Letter 0]

4 Dec 2020

JOURNAL REQUIREMENTS: 

Author response: The manuscript was adjusted to meet PLOS ONE’s formatting style requirements.

Author response: We used the service of Editage to edit our manuscript

3. In the ethics statement in the Methods section and online submission information, please specify the type of informed consent that was obtained from the participants (for instance, written or verbal, and if verbal, how it was documented and witnessed).

Author response: More information in Ethics statement was added, as follows: “…Participants were fully informed of the purpose of the study, the procedures involved, and of the commitment to keep their information confidential. Their written consent was obtained via an informed consent form that was attached to the survey cover letter. By clicking on the “Start survey” button on the Zoho Survey tool, participants indicated their consent and willingness to participate.”

4. As part of your revision, please complete and submit a copy of the COREQ Guidelines checklist, a document that aims to improve reporting of qualitative studies for purposes of post-publication data analysis and reproducibility: https://www.equator-network.org/reporting-guidelines/coreq/. Please include your completed checklist as a Supporting Information file. Note that if your paper is accepted for publication, this checklist will be published as part of your article.

Author response: Our study was not a qualitative study. The reason for this confusion was explained in the Question 6 of the Reviewer #2. 

5. Please include additional information regarding the survey or questionnaire used in the study and ensure that you have provided sufficient details that others could replicate the analyses. For instance, if you developed a questionnaire as part of this study and it is not under a copyright more restrictive than CC-BY, please include a copy, in both the original language and English, as Supporting Information, or include a citation if it has been published previously.

Author response: As mentioned in the manuscript, the questionnaire used for this study comprises 3 sections: (1) Demographic and work-related information; (2) Clinical pharmacy tasks and (3) Job satisfaction. The content in the second section was exactly extracted from the Circular No. 31/2012/TT-BYT. Regarding the third section, the whole content was developed based on five studies/materials (We mainly focused on Reference 14 and 17 because they were designed in Vietnamese, which could avoid translation errors. Please see details below). In more detail, we combined all aspects/content which were used in these materials and then eliminated repeated aspects/content. Then we made some changes to some content to be more suitable for this study participants, who are CPs because the five studies/materials mentioned above were not designed specifically for surveying CPs’ job satisfaction but for evaluating job satisfaction of overall healthcare professionals/hospital pharmacists. Thereafter, the draft version of the questionnaire entered the pilot phase with 30 CPs. 

We included a completed questionnaire in both Vietnamese and English as Supporting Information

Five studies/materials used:

13. American Society of Health-System Pharmacists. Exploring facets of job satisfaction among U.S. Hospital. American Journal of Health-System Pharmacy. 2007;53:301-657.

14. Vietnamese Ministry of Health. Decision: On Providing templates and guidance on surveying satisfaction of patients and healthcare professionals (Decision No.3869/QĐ-BYT). 2019.

15. Liu CS, White L. Key determinants of hospital pharmacy staff's job satisfaction. Research in Social and Administrative Pharmacy. 2011;7(1):51-63. doi: https://doi.org/10.1016/j.sapharm.2010.02.003.

16. Seston E, Hassell K, Ferguson J, Hann M. Exploring the relationship between pharmacists' job satisfaction, intention to quit the profession, and actual quitting. Research in Social and Administrative Pharmacy. 2009;5(2):121-32. doi: https://doi.org/10.1016/j.sapharm.2008.08.002.

17. Tran BK. Nghiên cứu thực trạng nguồn nhân lực dược bệnh viện và xác định nhu cầu dược sĩ tại các bệnh viện đa khoa tuyến tỉnh, huyện trong giai đoạn hiện nay (An analysis of hospital pharmacist workforce and the demand of pharmacist in general hospitals) [Master of Pharmacy Thesis]: Hanoi University of Pharmacy; 2015 

6. In the Methods, please discuss whether and how the questionnaire was validated and/or pre-tested. If these did not occur, please provide the rationale for not doing so.

Author response: The questionnaire was pre-tested with 30 CPs. Based on their feedbacks, we made some amendments to wording and question order to be more understandable

7. We note that you have indicated that data from this study are available upon request. PLOS only allows data to be available upon request if there are legal or ethical restrictions on sharing data publicly. For more information on unacceptable data access restrictions, please see http://journals.plos.org/plosone/s/data-availability#loc-unacceptable-data-access-restrictions.

Author response: The data set (SAV file which could be run with SPSS software 20.0) is available on Google Drive repository at: 

https://drive.google.com/file/d/1E_H0ysCg9gqSLuIZYNlxS9_TpLXglVMO/view?usp=sharing

8. PLOS requires an ORCID iD for the corresponding author in Editorial Manager on papers submitted after December 6th, 2016. Please ensure that you have an ORCID iD and that it is validated in Editorial Manager. To do this, go to ‘Update my Information’ (in the upper left-hand corner of the main menu), and click on the Fetch/Validate link next to the ORCID field. This will take you to the ORCID site and allow you to create a new iD or authenticate a pre-existing iD in Editorial Manager. Please see the following video for instructions on linking an ORCID iD to your Editorial Manager account: https://www.youtube.com/watch?v=_xcclfuvtxQ

Author response: ORCID iD was added

REVIEWER #1: 

SPECIFIC

1. In abstract: more details about the method of data collection is better to be given.

Author response: The methods summary in Abstract was adjusted to be clearer, which is as follows: “This was a cross-sectional study. Self-administered questionnaires were distributed to all CPs in all 128 hospitals at HCMC via an online survey tool from May to June 2020.”

2. In the introduction: the objectives of the study are not clear and the term “To summarize” is not acceptable as clear objective and cannot express the effort done by the researchers. I suggest to change it “To describe the status of clinical pharmacy profession implementation and analyze the key factors affecting job satisfaction of CPs at HCMC”

Author response: This sentence was adjusted into: “…For these reasons, this study was carried out to examine the current status of the clinical pharmacy profession and to analyze the key factors affecting job satisfaction of CPs.”

3. In the method; in line 102 written that data collection process went through four steps, while details of steps followed and figure 1 diagram shows 3 steps only. In factor analysis more explanation was needed regarding factor loading analysis and to justify why retained factor loading greater than 0.5 and eliminated less than that and on what base done that.

Author response: 

• The word ‘four’ was corrected into ‘three’. 

• Regarding factor analysis, it is a technique that is used to reduce a large number of variables into fewer numbers of factors, so that the research data is easier to work with. According to the Reference ‘Hair JF et al. Multivariate Data Analysis. CENGAGE INDIA (2018); 2018’, it is shown that in the measurement model, all of the factor loadings considered must be at least 0.5 and loadings with value ≥0.5 are very significant, that is the reason why we decided to set the minimum accepted factor loading at 0.5.

4. In Results: in table (2) the 7th general task the word “surprise” has no meaning and I think you may be meant to say “supervise”. Term “Colloquies” is not appropriate as a factor for job satisfaction and I suggest to change it “ Inter & Intra professional relationship”

Author response: In the 7th general task, the word ‘surprise’ was changed into ‘surprising’ because sometimes clinical pharmacists must report about drug use activities as unexpectedly required by the Board of Directors and the Drug and Treatment Council. For the term ‘Colloquies’, I think you may be meant to say ‘Colleagues’. We agree with your suggestion and changed the all the terms ‘Colleagues’ into ‘Inter & Intra professional relationships’

GENERAL 

English language needs to be revised.

Author response: Thank you very much for your remark, the English used was checked and all necessary corrections were done.

REVIEWER #2: 

SPECIFIC

1. LINE 26: In the background in the abstract is states “numerous difficulties” but specific difficulties are never elaborated in the main manuscript, so I recommend to change the sentence to be more neutral. The study is about both positive and negative aspects that impact work, therefore I suggest “numerous factors” as a preferable framing in the background.

Author response: Thank you for your remark. We changed ‘numerous difficulties’ into ‘numerous challenges’.

2. LINE 42-43: States “favorable outcomes” and “limitations and barriers” I recommend to change this sentence that is more specific, with whichever the authors think are most noteworthy from the main conclusions.

Author response: The mentioned sentence was re-written into “Most clinical pharmacy tasks noted a high rate of participation from the CPs. Nevertheless, hospitals in HCMC was found to be experiencing a shortage of CPs and low levels of participation of CPs in ward rounds, and most CPs were unable to completely focus on clinical pharmacy tasks”

3. LINE 50: It would be helpful for a reader unfamiliar with Vietnam policy (and also unable to locate Vietnam legal records) that the authors consider to add a brief description about the first policy including some key roles/tasks for CPs that are in this policy. In addition, as a reader I am interested to know what was the main rationale or cause for this policy to be created. For example, was there a growing rate of medication errors that caused this? Was it because of request from the hospitals? Was the policy a result of the work of professional advocates?

Author response: 

• In this policy, there were a total of 14 tasks that CPs have to handle. These tasks were elaborately mentioned in Table 2 of the Result section and thus, we decided not to include some of them in the Introduction section. 

• To clarify the reason behind the creation of the first legislation about clinical pharmacy (Circular No. 31/2012/TT-BYT), more information was added into and some adjustments were made in this paragraph, as follows: “…It (Clinical pharmacy) has been a common and core practice within healthcare systems in developed countries since the 1960s [2, 3]. However, it was not until 2012 that the Vietnamese Ministry of Health (MOH) imposed the first ever regulation to determine and elaborate on the role and tasks of a clinical pharmacist (CP) (Circular No. 31/2012/TT-BYT). This legal document came into existence because clinical pharmacy was into being practiced at a few top-ranked hospitals via international collaborative programs, and not at many other small and medium-sized hospitals. Thus, there was a necessity to promulgate an official legislation to guide and to oblige all hospitals in Vietnam to gradually implement clinical pharmacy services in their hospitals.”

4. LINE 54: “barriers” and “limitations” that are described as being a problem for CPs. I understand that the authors refer to previous reports and previous research. However, given that these ideas are foundational to the present investigation, a brief explanation as to the nature of the problems would be helpful. This will also help assessment the alignment between the foundation of the study, the chosen measures and the conclusions. A simple sentence that states, “For example…..” would likely help orient the reader.

Author response: Some ‘barriers’ and ‘limitations’ we wanted to mention were added as recommended. The referred sentence was re-written into “…barriers and limitations, such as shortage of workforce, the lack of qualified clinical pharmacists (CPs), and the lack of interaction between CPs and other healthcare professionals”

5. LINE 66-68 This section is talking about the important of assessing job satisfaction. To help make this claim more credible, it is recommend to add a citation of previous research that makes the link between job satisfaction and quality of work or job performance.

Author response: References which point out the impact of job satisfaction on job performance were added, as follows:

• Bond CA, Raehl CL. Pharmacists' Assessment of Dispensing Errors: Risk Factors, Practice Sites, Professional Functions, and Satisfaction. Pharmacotherapy: The Journal of Human Pharmacology and Drug Therapy. 2001;21(5):614-26. doi: https://doi.org/10.1592/phco.21.6.614.34544.

• Gidman WK, Hassell K, Day J, Payne K. The impact of increasing workloads and role expansion on female community pharmacists in the United Kingdom. Research in Social and Administrative Pharmacy. 2007;3(3):285-302. doi: https://doi.org/10.1016/j.sapharm.2006.10.003.

• James KL, Barlow D, McArtney R, Hiom S, Roberts D, Whittlesea C. Incidence, type and causes of dispensing errors: A review of the literature. International Journal of Pharmacy Practice. 2009;17(1):9-30. doi: https://doi.org/10.1211/ijpp.17.1.0004.

6. LINE 75: The design mentions qualitative, but I don’t recall the results of this part in the results section. Can the authors clarify what aspects were evaluated for each of the qualitative and quantitative sections from the study?

Author response: Thank you very much for your remark. Initially, we aimed to conduct two separate studies on clinical pharmacists’ job satisfaction and job stress, wherein the job stress study was the one combining qualitative and quantitative analysis and this study was a merely quantitative study. The manuscripts for these two studies were written nearly at the same time, which resulted in the small confusion for us. Thanks to your remark, we made a correction in Study design section.

7. LINE 88-90: It is not clear what the tasks are that are called “tasks concurrently handled” compared to “clinical pharmacy tasks” and “other traditional pharmacy practice.” Clinical pharmacy tasks are specific to the new law, but it is recommended that more clear examples to help distinguish what it means by a “task concurrently handled” means. For example, is this non-pharmacy tasks like cleaning or something else?

Author response: In Vietnam, human resources of hospital pharmacy practice are divided into 6 branches, which include: 1) Professional pharmacists; 2) Pharmacists in charge of the storage facility and provision of drugs; 3) Pharmaceutical statisticians; 4) Pharmacists who prepare, test and control the quality of drugs; 5) Pharmacists who manage the specialized activities of the Hospital Pharmacy and 6) Clinical pharmacists. Among 6 branches, clinical pharmacists branch is the newest. Therefore, duties handled by the 5 remaining branches of pharmacists are also known as traditional pharmacy practice. 

In closing, there are a total of six groups of tasks (six duties) available in hospitals in Vietnam, which are handled by above branches of pharmacists. However, due to the lack of workforce, pharmacists sometimes must join in other groups of tasks as well besides the group that they are mainly responsible for. In order to clarify, we would replace ‘number of tasks concurrently handled” with the phrase ‘number of duties that CPs handle from among 6 duties (professional pharmacy, storage and provision of drugs, pharmaceutical statistics, controlling the quality of drugs, managing the specialized activities of the hospital Pharmacy, and clinical pharmacy)’. We made some similar adjustments in other sections. 

8. Line 91: It is unclear why “time spend on ward round participation” is separate from “clinical pharmacy.” It seems these are both in the same category of ideas that a CP should do. Explain why this task is separated out from the rest.

Author response: Ward round is a regular visit to patients in hospital by medical staff(s) (alone or with a team of healthcare professionals) for the purpose of making decisions concerning patient care. Basically, ward round is part of clinical pharmacy. In developed countries, ward round participation by a CP is very common in hospitals. As far as we know, in Vietnam in general and in HCMC in particular, ward round has been already implemented in top-ranked hospitals, but still at a small-scale. That is the reason why we want to know the status of implementation of ward round in all hospitals at HCMC via the number of CPs perform ward rounds and the amount of time they spend on this activity.

9. Line 159: Similar to previous comments, it is unclear what is meant by “concurrently handle various tasks” Some examples in the methods or results section would help the reader understand what these mean. Does this mean they have 2 different jobs?

Author response: The answer for this question was already mentioned in the Question 7 (Line 88-90) 

10. Figure 2 needs more information in the figure so that the figure can stand-alone for interpretation. As to previous comments, it is not clear how on-ward participation is different from clinical pharmacy. Are these in the same category? Also, does the middle bar (about “other” duties include “tasks concurrently handled” as well as traditional pharmacy services? I think if these categories were better described in the methods, then in the results would be more clear. Also the figure need to give a title.

Author response: Figure 2 was adjusted to become more understandable. The title for Figure 2 is “Mean time that clinical pharmacists (CPs) spent on clinical pharmacy duty, traditional pharmacy duties, and clinical ward rounds [Mean (Standard deviation)]”.

11. LINE 170-181: I am not sure what on ward rounds is not included in this assessment of clinical pharmacy tasks. However, the clinical pharmacy tasks appear to be slip between on and off the ward duties. I think this confusion can be clarified with the definitions and rationale for these categories within the methods section. Then, the results will be more understandable.

Author response: The division of clinical pharmacy tasks into general activities and activities in the clinical wards in this study was totally based on the Circular 31/2012/TT-BYT and we already mentioned this in the Result section.

12. LINE 247: This sentence and ideas about how the PhD Pharm degree is different needs to be re-written for clarity. It is unclear how this is different or similar to the PharmD degree. “…Focus more on practice aspect” needs to be clarified.

Author response: The above sentence was re-written into “…Meanwhile, many countries all over the world, such as the United States, Canada, Korea, Japan, Pakistan, Thailand, etc. have been gradually eliminating the B. Pharm degree and making it mandatory for a pharmacist to possess a Pharm. D (Doctor of Pharmacy) degree to participate in clinical pharmacy activities (the Pharm. D is a doctorate degree similar to Ph. D. Pharm, except that it is a professional or a more clinically oriented degree, whereas a Ph. D. Pharm is a research graduate degree)”

13. LINE 249: A stronger case will need to be made of the rationale of the need to increase credentialing for CPs. If they are currently fulfilling the duties as required, what evidence can be provided to correlate increased credential to increase workforce quality?

Author response: In order to clarify the necessity to make a transition from B. Pharm to Pharm. D, more information about B. Pharm education was added, as follows: “…Further, as per a publication on pharmacy education in Vietnam, the B. Pharm degree in general is more product-oriented and mainly focuses on laboratory-based courses, while clinical training and practical experience are not given adequate attention. As a result, pharmacy graduates in Vietnam tend to excel more in the pharmaceutical industry, such as drug manufacturing, research, quality assurance and control, and drug discovery, but when it comes to clinical pharmacy practice, most of them are not equipped with sufficient expertise and hands-on experience. This weakness, however, can be entirely surmounted with introducing a Pharm. D curriculum. Therefore, administrative agencies should consider creating a clear pathway for the transformation of CPs’ academic degrees from B. Pharm to Pharm. D, in terms of both clinical pharmacy education and practice, in order to achieve the long-term objectives of raising the quality of the clinical pharmacy workforce and catching up with the rest of the world.”

14. LINE 255: The authors compare full-time CPs with others who have the handle “at least 2 tasks” – this makes me believe that this mean like jobs different from pharmacy. However, see earlier comments about clarifying what this work category means.

Author response: You can see above answers to understand these terms more clearly. In more detail, we used the term ‘full-time CPs’ to refer to CPs who only handle clinical pharmacy duty, whereas ‘CPs who handle at least 2 duties’ means that they have to handle clinical pharmacy plus at least one traditional pharmacy duty (the amount of time they spend on clinical pharmacy would reduce in order for them to handle other traditional pharmacy duties).

15. LINE: 299: The idea that executives, and internal regulations impact job satisfaction is a very interesting finding. Therefore, I believe the authors should discuss potential ways that these policies do to impact CPs. Why do the authors believe this was the biggest factor?

Author response: We have added the content of the discussion that you commented on. Sincerely thank you for your contributions. Regarding your question, we have the following answers: 

(1) clinical pharmacy is a job that requires a lot of effort and great work pressure. At the same time, we also point out in our research that in addition to clinical pharmacy, clinical pharmacists have to perform many other tasks. However, the clinical pharmacist's income was in proportion. Therefore, internal regulations such as creating conditions for clinical pharmacists to have more study, promotion opportunities, and more comfort at work, ... will have a great impact on job satisfaction. 

(2) The relationship between leadership style and job satisfaction of health workers has been demonstrated in studies around the world. In Vietnam, when the role of an executive is always at the center, the executives and the internal regulations established by them exert great influence on employees’ trust and their level of job satisfaction. This is the basis for leaders to consider having a long-term leadership vision, as well as to develop and expand suitable internal regulations. Since then, the satisfaction level of CPs at the hospital were found to have been increasing.

GENERAL

Recommend remove the use of the term “prove” as this is too definitive for being used to refer to ongoing study or only one or a few research studies.

Author response: We changed the term ‘prove’ into ‘show’ based on your recommendation.

---

## [Decision Letter · Decision Letter 1]

2 Jan 2021

Job satisfaction of clinical pharmacists and clinical pharmacy activities implemented at Ho Chi Minh City, Vietnam

PONE-D-20-31080R1

Dear Dr. Hai-Yen Nguyen-Thi,

We’re pleased to inform you that your manuscript has been judged scientifically suitable for publication and will be formally accepted for publication once it meets all outstanding technical requirements.

Kind regards,

John Rovers, PharmD, MIPH

Academic Editor

PLOS ONE

Additional Editor Comments (optional):

Reviewers' comments:

Reviewer's Responses to Questions

**Comments to the Author**

1. If the authors have adequately addressed your comments raised in a previous round of review and you feel that this manuscript is now acceptable for publication, you may indicate that here to bypass the “Comments to the Author” section, enter your conflict of interest statement in the “Confidential to Editor” section, and submit your "Accept" recommendation.

Reviewer #2: All comments have been addressed

2. Is the manuscript technically sound, and do the data support the conclusions?

Reviewer #2: (No Response)

3. Has the statistical analysis been performed appropriately and rigorously? 

Reviewer #2: (No Response)

4. Have the authors made all data underlying the findings in their manuscript fully available?

Reviewer #2: (No Response)

5. Is the manuscript presented in an intelligible fashion and written in standard English?

Reviewer #2: (No Response)

6. Review Comments to the Author

Reviewer #2: (No Response)

7. PLOS authors have the option to publish the peer review history of their article (what does this mean?). If published, this will include your full peer review and any attached files.

Reviewer #2: No

---

## [Editor Report · Acceptance letter]

13 Jan 2021

PONE-D-20-31080R1 

Job satisfaction of clinical pharmacists and clinical pharmacy activities implemented at Ho Chi Minh city, Vietnam 

Dear Dr. Nguyen-Thi:

I'm pleased to inform you that your manuscript has been deemed suitable for publication in PLOS ONE. Congratulations! Your manuscript is now with our production department. 

Kind regards, 

on behalf of

Dr. John Rovers 

Academic Editor

PLOS ONE